# The Moderator Effect of Subthreshold Autistic Traits on the Relationship between Quality of Life and Internet Addiction

**DOI:** 10.3390/healthcare11020186

**Published:** 2023-01-07

**Authors:** Francesco Sulla, Michela Camia, Maristella Scorza, Sara Giovagnoli, Roberto Padovani, Erika Benassi

**Affiliations:** 1Department of Human Studies, University of Foggia, 71121 Foggia, Italy; 2Department of Biomedical, Metabolic and Neural Sciences, University of Modena and Reggio Emilia, 41125 Modena, Italy; 3Department of Psychology “Renzo Canestrari”, University of Bologna, 40127 Bologna, Italy; 4Child and Adolescent Neuropsychiatry Service, AUSL Modena, 41124 Modena, Italy

**Keywords:** sub-threshold autistic traits, quality of life, Internet addiction, problematic Internet use

## Abstract

People with sub-threshold autistic traits (SATs) are more prone to develop addictive behaviors such the ones linked to Internet abuse. The restrictions for anti-COVID-19 distancing measures encourage social isolation and, consequently, increase screen time, which may lead to Internet addiction (IA). However, a better quality of life (QoL) may have function as a protective factor against the development of IA. This study wanted to investigate the relation between SATs, QoL, and the overuse of the Internet in a group of 141 university students in the North of Italy. Participants completed a battery of tests. Results suggest that QoL is a predictive factor of IA and that the relationship between QoL and IA is significantly moderated by SATs. This could mean that SATs might represent a risk factor for IA, even when people have a better quality of life. Differences between female and male students are discussed, as well as possible implications for practice.

## 1. Introduction

Several studies show that features of autism are not limited to individuals that have a clinical diagnosis, and that autism-like traits, as in a continuous view, vary throughout the general population at lower severities, and are expressed as sub-threshold autistic traits (SATs) e.g., [1,2,3]. SATs in adults from the general population “reflect similar, though less severe, social-cognitive and emotional features compared to those observed in autism spectrum disorders” (ASD) [4] (p. 187). This mean that the impairments present in ASD may be distributed continuously through the population. That is, “differences between people with an autism spectrum diagnosis and people with sub-threshold behavioural traits are quantitative rather than qualitative in nature” [5] (p. 4065). In addition, empirical data suggest that higher levels of SATs in young adults are linked to a lower number and duration of friendships, increased loneliness [6], higher depression, anxiety, and difficulties in social and personal adjustment [7]. Moreover, some of these traits might be responsible for the increase in the risk of developing addictive behaviors. Indeed, the core symptoms of ASD are seen to be related, for example, to the development of problematic video game use [8,9,10,11]. Furthermore, recent evidence suggests that people with ASD having average or above-average intelligence quotients—which is the case for people expressing SATs—are more likely to develop addictive behaviors such the ones linked to Internet abuse [12]. As a confirmation of this, Hirota et al. [13] conducted a longitudinal study lasting 2 years that studied the patterns of internet use in elementary and junior high school pupils. They found that, in this population, autistic traits predict Internet addiction (IA). However, neurodevelopmental traits were measured only once at baseline, precluding analyses on longitudinal interactions between those traits and IA issues.

The pandemic situation we are still in the middle of might have had an impact on some problematic behaviors. Indeed, a recent systematic review [14] proved an increase in Internet-based addictive behaviors during the pandemic period. In fact, the restrictions for anti-COVID-19 distancing measures encourage social isolation and, consequently, increase screen time, which may lead to IA, e.g., [15,16]. The review revealed that, during the pandemic, there was a significant increase in Internet usage of about 52% and people mostly utilized mobile phones for accessing the Internet. Although official diagnostic criteria do not exist yet, IA can be identified as “the excessive, obsessive-compulsive, uncontrollable, tolerance-causing use of the Internet, which also causes significant distress and impairments in daily functioning” [17] (p. 5). IA is not included in the latest Diagnostic and Statistical Manual of Mental Disorders (DSM-V). It has been mentioned as a disorder that requires further studies [18]. Internet use may be considered problematic “when an individual experiences a maladaptive preoccupation and a sense of craving for online activities and remains connected longer than intended, despite the negative consequences (distress or impairment) resulting from that behavior” [19] (p. 3).

There is a consensus in the literature that negative overall Internet use is associated with psychosocial problems, such as higher levels of depression, increased social anxiety and loneliness, poorer psychosocial well-being [13,20,21,22,23], and health-related quality of life (QoL) [24,25,26,27]. Health-related QoL (HRQoL) has been extensively studied in adult population in numerous medical pathologies [28], while there is a paucity of studies on populations such as university students. What is known is that, during childhood–adolescence, girls are twice as likely as boys to perceive worse health, as well as worse QoL, especially in the physical and emotional dimensions [29]. Seeing that (also in Italy) a persistent under-representation of women in science, technology, engineering and mathematics (STEM) areas (e.g., [30]) exists, a difference in these dimensions of QoL might be also found in students enrolled in scientific/economic-related majors and students enrolled in humanities-related majors. Moreover, while there is evidence from several studies conducted before the pandemic in which students from the health sciences or engineering areas were found to present higher symptomatology scores than those in the humanities area (e.g., [31,32]), the results of a recent study [33] conducted during the pandemic show exactly the opposite. In fact, the lowest scores in QoL were shown by engineering and architecture students. In line with their results, the study of Lipson et al. [34] shows that “Arts and Humanities students have a greater tendency to develop mental illnesses compared to the other areas, such as the Engineering and Business students, who also seem to undergo treatment less frequently” ([33] p. 113108). Hence, differences in QoL of students enrolled in different majors should be further investigated.

However, some domains of QoL are seen to be protective factors for Internet addiction, and this is even more applicable for younger age groups [35].

Individuals quarantined at home during the pandemic period reported higher anxiety and depressive symptoms and lower QoL levels (e.g., [36]). Looking at the literature just analyzed, this, in turns might have had an impact on the risk of IA, especially in individuals considered vulnerable, such as young adults with sub-threshold autistic traits.

### Present Study

As confirmed by a recent cumulative review of the literature on factors that predispose undergraduates to mental issues, technology and other new addictions in this population are still little studied [37]. A few examples of correlational studies were found that investigated the relationship between quality of life and IA: for example, Fatehi et al. [38] found negative correlation between IA and QoL in a group of Iranian medical students; the same results were found by Chern and Huang [39] in a large sample of college students in Taiwan. Moreover, the results of a study by Dell’Osso et al. [40] found that university students with problematic Internet use show higher levels of autistic traits compared to those without problematic Internet use. However, to the best of our knowledge, no studies investigated the relationship between these three variables together in a population of university students.

We hypothesis that QoL has a different impact on IA depending on the presence or absence of SATs. In people with lower/no SATs, QoL might represent a protective factor against the risk of developing IA. The same might not be true for people with high SATs. This might be due to the fact that at combination of SATs and low QoL could result in a closure towards the external world, both real and virtual—resulting in a less frequent use of the Internet and less risk of developing IA. On the other hand, high SATs together with better QoL might result in a greater push towards socialization—mainly virtual in case of people with SATs—and a consequent increase in Internet use in this population.

Therefore, the main goal of the present study was to investigate the relation between subthreshold autistic traits, quality of life, and the overuse of the Internet among university students. Considering the abovementioned theoretical premises, we hypothesized a decrease in IA risk as quality of life increases, with a moderating effect of SATs (Figure 1).

## 2. Materials and Methods

### 2.1. Participants

This study involved a convenient sample of 141 undergraduate university students, enrolled at University of Modena and Reggio Emilia (Italy). Among the study sample, 106 (75.2%) were female and 35 (24.8%) were males. Their mean age was 23.26 years (SD = 2.61, range 19–31). The students attended various academic degree courses identified as follow: 74 students (52.5%) in the scientific/economics area, 67 students (47.5%) in the humanities area. All students had no history of major cerebral damage, congenital malformations, visual and hearing impairments, neuropsychiatric disorder, and educational deficits.

All the participants for this study were recruited in the university campus in the months of April–May 2021, i.e., just over a year after the beginning of the COVID-19 pandemic.

### 2.2. Procedure

Students were invited to participate in the study by email or mobile phone, and were informed in detail about the aims of the study, the voluntary nature of their participation, and their right to withdraw from the study at any time. Participants provided their informed written consent for participation in the study, data analysis, and data publication, and independently completed a self-administered and structured on line questionnaire, which took around 20 min to complete. Specifically, for testing QoL, IA and the presence of subthreshold autistic traits three standardized questionnaires were used, namely: the PedsQL^TM^ 4.0 generic core scales young adult version, the Internet Addiction Test (IAT), and the Autism Spectrum Quotient (AQ).

### 2.3. Measures

PedsQL^TM^ 4.0 generic core scales young adult version [41]. The PedsQL^TM^ measures the core dimensions of quality of life (QoL), as delineated by the World Health Organization, in young adults aged 18–25. Respondents are asked to answer 23 items referring to problems during the past month. Responses are measured on a five-point Likert-type scale ranging from 0 to 4 (0 = never a problem, 4 = almost always a problem). Items are reversed scored and linearly transformed to a 0-100 scale as follows: 0 = 100, 1 = 75, 2 = 50, 3 = 25, 4 = 0. Hence, rough item responses are rescaled into a range from 0 to 100, with 0 meaning the worst and 100 meaning the best levels of QoL. The items are grouped into 4 multidimensional subscales: physical functioning (QoL P; 8 items), emotional functioning (QoL E; 5 items), social functioning (QoL S; 5 items), and school/work activities (QoL W; 5 items). These four subscales could be grouped into three composite scores: a total score (23 items) (QoL G), a physical health summary score (QoL P), and a psychosocial health summary score (QoL E + QoL S + QoL W). The PedsQL^TM^ demonstrates good psychometric properties. Internal consistency reliabilities (Cronbach’s alpha) exceed the standard of .70 for group comparisons. Across the ages, the total scale score for self-report and proxy-report approaches an alpha of .90 [42].

Internet Addiction Test (IAT) [43,44]. The first version of the IAT was developed by Young (1998) and was based on the DSM-IV criteria for pathological gambling (i.e., tolerance, withdrawal symptoms, mood modification or relapse). According to the author [43], the IAT was developed to measure the subjective risk of using the Internet (e.g., trying to hide the time spent online) and gives an account of the degree to which technological abuse affects daily routine, social life, work and school productivity, and even sleep quality. The IAT is a self-report 20-item questionnaire with each item rated on a five-point Likert-type scale ranging from 0 (not applicable) to 5 (always). According to the Italian validation [44], a total score from 0 to 30 represents average users with complete control of their Internet use; a total score from 31 to 49 represents a mild level of IA; a score from 50 to 79 represents moderately addicted users; a score from 80 to 100 represents severely addicted users. The scale demonstrates very good internal consistency and a strong internal reliability across studies (e.g., [45] reported an alpha of 90).

Autism Spectrum Quotient (AQ) [46,47]. The AQ is a self-report questionnaire designed to identify autistic traits in the general population. A total of 50 questions assess autistic traits across five areas (10 questions per domain), including: (1) social skills; (2) communication; (3) attention to detail; (4) attention switching; and (5) imagination. Responses are rated on a four-point Likert-type scale ranging from ‘definitely agree’ to ‘definitely disagree’. Half of the questions are reverse-coded. It utilizes a binary scoring method where the presence of autistic traits, either mild or strong, is scored as a +1, while the opposite is scored as 0. The total score can range between 0 and 50. Higher scores indicate more autistic traits. Scores of 32 or above may be suggestive of ASD. The AQ shows consistent results across time and across different age groups in independent samples (e.g., [48,49]), and has good cross-cultural stability. The internal consistency of items in each of the five domains are all moderate to high (communication α = 0.65; social, α = 0.77; imagination α = 0.65; local details α = 0.63; attention switching α = 0.67) (e.g., [50]).

### 2.4. Data Analyses

All statistical analyses were carried out using SPSS 27.0 for Windows. Tests were bilateral with a statistical significance set at 0.05. Preliminary analyses of data distribution (Kolmogorov–Smirnov test, skewness and kurtosis) show that most of the study’s variables, including total score of the target variables, are not normally distributed. Therefore, non-parametric analyses are applied.

A set of Mann-Whitney tests were preliminary conducted to verify differences between males and females and between students enrolled in a scientific/economics major and students enrolled in a humanities major in the study’s variables (i.e., QoL, Internet addiction, SATs).

Preliminary Spearman’s correlational analyses were performed in order to test associations between quality of life (i.e., PedsQL^TM^ subscales), autistic traits (i.e., AQ), and Internet addiction measure (i.e., IAT). Finally, a moderation model was performed using the Generalized Linear Model (GLM). This model assumes that the dependent variable is linearly related to the factors and covariates via a specified link function. Moreover, the model allows for the dependent variable to have a non-normal distribution, as in the case of the variables used in this study. The statistical assumption of the GLM (i.e., statistical independence of observations, correct specification of link function) has been carefully checked and criteria were met.

Moderation analysis was conducted to analyze the effect of QoL on IA (i.e., IAT scores) moderated by SATs (i.e., AQ scores). QoL was used as predictor of IA, and SAT was set as moderator factor. To perform the moderation models by means of GLM, the main effects of the QoL and SATs, and the interaction effects of QoL by SATs (moderation effect) were analyzed.

## 3. Results

### 3.1. Descriptive and Correlational Analyses

Table 1 presents the descriptive statistics of key variables by gender. The results of the Mann-Whitney U tests reveal significant differences between males and females in the physical functioning, with the female university students reporting lower levels than males, i.e., a worse condition on these QoL dimensions. There are no significant differences between males and females for the emotional, social, school/work dimensions, or for the QoL total score. No significant differences emerge between males and females for Internet addiction and presence of autistic traits either.

A comparison between students enrolled in a scientific/economics major and students enrolled in a humanities major is presented in Table 2. Significant differences are found in both physical and emotional dimension of QoL and in QoL total score. In particular, students enrolled in humanities majors report lower scores, showing a significant lower quality of life in the aforementioned dimension of PedsQL^TM^. No significant differences are found between the two groups regarding the other QoL dimensions. The Mann–Whitney test does not reveal significant differences between the two groups of students for Internet addiction and presence of sub-threshold autistic traits.

As shown in Table 3, Spearman’s correlations reveal that all the QoL dimensions are negatively associated with IA. Moreover, the presence of sub-threshold autistic traits is positively associated with IA.

### 3.2. The Moderation Effect of SAT on the Relationship between QoL and IA

Attesting to the fact that the dependent variable (IA) is not normally distributed, in order to evaluate the moderator role of SAT on the relationship between QoL traits and IA, a moderation analysis by the means of the GLM was performed. The model returns a significant level (Likelihood Ratio Chi-Square = 65.12, *p* < 0.01), attesting to the fact that the factors considered are, when taken together, predictive for IAT.

The results show a significant interaction effect between QoL and SAT (moderator effect: Wald Chi-Square = 5.28, *p* = 0.02), attesting that QoL results are predictive for IA when moderated by SAT. The main effect of QoL on IA is significant (Wald Chi-Square = 19.43, *p* < 0.001) whereas the main effect of SAT on IA is not significant (Wald Chi-Square = 3.21, *p* = 0.071). These results suggest that QoL is a predictive factor of IA and that the relationship between QoL and IA is significantly moderated by SATs (Table 4).

For descriptive purposes, we explored the nature of the interaction between QoL and SAT using a simple slope analyses. The simple slope was obtained by calculating predicted values of IA under different QoL and SATs conditions. To create the simple slope, a calculation of values corresponding at the mean, one standard deviation above and below the mean for both predictor (QoL) and moderator (SAT) was performed. These values were used to plot the variables and to test the statistical significance for each of the simple slopes. In Figure 2 the relation between QoL and IA at different levels of SATs (i.e., −1SD, mean and +1SD) is shown.

The results suggest that the QoL predicts IA at all the different levels of SAT. In detail, IA scores increase with the increasing QoL scores in all the SATs level. The strongest effect is visible in the low-SAT group (one standard deviation below the mean; b = −0.036, LLCI-ULCI = −0.459–−0.262). Similar effects are found in the high-SAT group (one standard deviation above the mean; b = −0.021, LLCI-ULCI = −0.029–−0.013) and in the mean-AQ group (b = −0.028 LLCI-ULCI = −0.035–−0.022) (Figure 2).

## 4. Discussion

The present study wanted to investigate the relation between subthreshold autistic traits, quality of life, and the overuse of the Internet in a group of university students.

Similarly to previous studies (e.g., [29]), female participants are found to suffer a greater loss of physical QoL than males; unlike the cited research, participants in the current study do not differ in the emotional dimension of QoL. It is established in the literature that feelings of fatigue and low energy compromise quality of life in college students and are influenced by sleep quality, sedentary time, and physical activity in an interactive manner and differentially by sex. For example, a recent study [51] found differences between male and female college students, with sleep quality for males and physical activity for females being more closely related to energy and fatigue outcomes. Moreover, a possible explanation may reside in the fact that, since the advent of COVID-19 vaccinations, many girls and women around the world are reporting abnormalities in their menstruation post-vaccination. Indeed, the literature in the field reports several menstrual abnormalities following COVID-19 vaccination, including increased cycle length, pain, and bleeding [52]; a study [53] reported a possible link between the vaccine for COVID-19 and menstrual abnormalities that have impacted girls and women’s QoL. However, seeing the cross-sectional nature of this investigation, one cannot be sure if the difference in the physical dimension of QoL in this sample was pre-existing or specifically due to conditions that emerged during the pandemic.

Differences in QoL are also found between students enrolled in scientific/economics (included into STEM areas) majors and students enrolled in humanities majors. Differences might not be related to gender as hypothesized, seeing that, in this case, there are also differences in emotional area and in the general score for QoL. Students enrolled in scientific/economics majors have better results in the aforementioned dimensions similar to other studies conducted pre pandemic [31,32], in contrast to results found in Spain during the pandemic [33]. Differences might be due to the realisation that STEM disciplines offer the highest paid and most stable career paths—a trend that is destined to strengthen in a world increasingly dependent on technologies (e.g., [54]). This results in a rise in new research questions that need to be further investigated in future research.

The results of the study confirm the main hypothesis, finding that, as the scores in quality of life increase and the scores in autistic trait decrease, Internet addiction scores decreases. This relationship is evident at all levels of autistic traits. In particular, quality of life appears to have a greater effect on internet addiction when autistic traits are lower, meaning that, in people with SATs this may be less protective in regards to the risk of IA.

A better quality of life can be a protective factor for the development of Internet addiction. Subthreshold autistic traits, on the other hand, might represent a risk factor for Internet addiction, even when people have a better quality of life.

It must be said, however, that people with autism may use the Internet in qualitatively different ways from those without autism [55]. Studies on Internet use by adults with autism show that information and communication technology “is important to them, to forge their identity and find a place in which they belong to a social network” [56] (p. 2). In a study by Shpigelman and Gill [57] the majority of participants reported finding it easier to make new friends using the Internet and being more comfortable communicating online than face-to-face. Moreover, adults with autism say that “the Internet provides them with a (cyber)space where they can escape their parents’ or caseworkers’ control and be more self-determined” [56] (p. 2).

In general, Internet use has become a necessary component of daily life as it continues to satisfy our requirements for information, entertainment, and social interaction. This is especially true for university students, whose learning and social networking both involve the use of the Internet. Undoubtedly, a balance needs to be maintained between Internet use and misuse, especially in a population with sub-threshold autistic traits.

Currently, in Italy, no services aimed at IA screening, prevention, or intervention exist either in universities or within the national health service. This might be due to the fact that official diagnostic criteria do not exist yet [17]. However, this has to be considered an issue of concern for health professionals and society in general, as it is common among clinical and non-clinical populations (as people showing SATs [40]) and it is well-known that problematic Internet use may lead to various difficulties in domains such as mental attitude, eating and sleeping habits, preventive behaviors, and health practices [13,20,21,23,24]. University counseling services might have a key role, designing awareness campaigns and prevention programs. Furthermore, student counseling professionals should pay more attention to this issue and try to provide opportune assistance to people exhibiting early manifestations of IA. On the other hand, awareness should be raised among health professionals regarding the fact that features of autism are not limited to individuals that have a clinical diagnosis, and they may be trained to assess for SATs.

Nevertheless, while this study systematically examines the relationship between autistic traits, quality of life, and IA, caution needs to be used when interpreting the causality, temporality, and generalizability of the results because of the cross-sectional design of this study, the self-reporting nature of the data, and the relatively small sample size, consisting of mostly female students. Moreover, data on ethnicity of the participants should be collected in future studies in order to assure a more diverse sample. More empirical research is also warranted to further explore the relationships between the investigated variables as found in this study and to help develop effective IA prevention programs.

## Figures and Tables

**Figure 1 healthcare-11-00186-f001:**
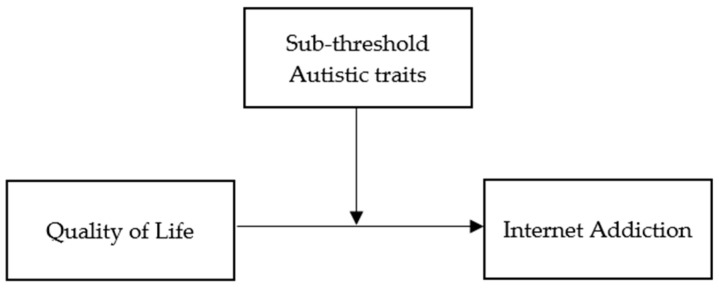
Hypothesized model.

**Figure 2 healthcare-11-00186-f002:**
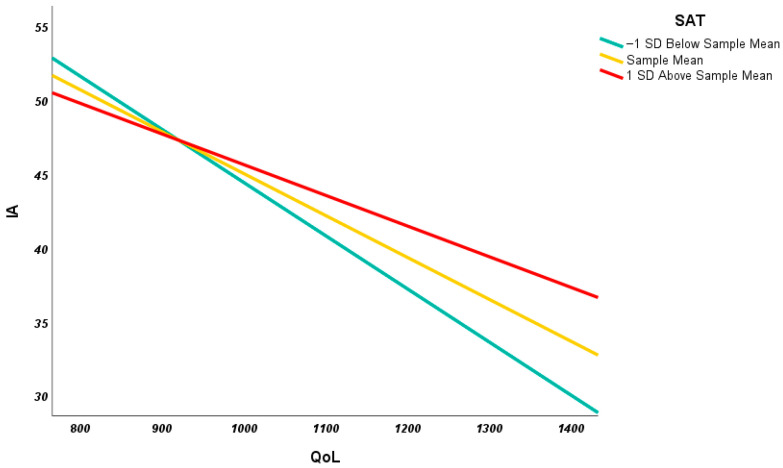
Relationship between QoL and IA at the high, average and low levels of SATs.

**Table 1 healthcare-11-00186-t001:** Descriptive data for quality of life (PedsQL^TM^), internet addiction (IAT), and sub-threshold autistic traits (AQ) by gender, and results of the statistical comparisons (Mann-Whitney tests). Significant results are reported in bold.

		Male Students (*n* = 35)		Female Students (*n* = 106)			
Variables		Mean (SD)	Range	Mean (SD)	Range	U	*p*
Quality of life (PedsQL^TM^)	Physical functioning	615.03 (138.47)	201–775	572.41 (131.97)	100–800	1365.00	0.019
	Emotional functioning	291.43 (112.78)	75–500	244.81 (116.89)	0–450	1467.00	0.063
	Social functioning	74.14 (34.18)	2–102	83.13 (23.35)	27–102	1661.00	0.315
	School/work activities	140.71 (36.92)	50–200	144.81 (35.64)	50–200	1772.50	0.687
	Total score	1121.31 (266.24)	428–1527	1045.16 (237.54)	327–1527	1472.00	0.067
Internet addiction (IAT)		42.00 (9.29)	24–57	42.72 (11.66)	21–70	1817.00	0.858
Sub-threshold autistic traits (AQ)		18.89 (5.17)	10–30	18.63 (6.27)	7–34	1772.00	0.692

**Table 2 healthcare-11-00186-t002:** Descriptive data for quality of life (PedsQL^TM^), Internet addiction (IAT) and sub-threshold autistic traits (AQ) by major area, and results of the statistical comparisons (Mann–Whitney tests). Significant results are reported in bold.

		Students Studying in Sciences/Economics (*n* = 65)		Students Studying in Humanities (*n* = 76)			
Variables		Mean (SD)	Range	Mean (SD)	Range	U	*p*
Quality of life (PedsQL^TM^)	Physical functioning	623.86 (122.90)	200–775	556.58 (138.89)	10–800	1800.50	**0.005**
	Emotional functioning	282.69 (113.79)	50–500	233.88 (116.17)	0–450	1910.50	**0.020**
	Social functioning	81.23 (28.84)	2–102	80.62 (24.73)	27–102	2339.00	0.557
	School/work activities	144.23 (31.80)	50–200	143.42 (39.24)	50–200	2465.50	0.985
	Total score	1122.02 (236.14)	428–1527	1041.50 (245.31)	327–1402	1854.00	**0.011**
Internet addiction (IAT)		41.63 (9.85)	24–57	43.32 (12.07)	21–70	2337.50	0.583
Sub-threshold autistic traits (AQ)		18.75 (5.39)	10–30	18.64 (6.51)	7–34	2384.50	0.723

**Table 3 healthcare-11-00186-t003:** Spearman’s correlations (rs) between quality of life (PedsQL^TM^ subscales) and Internet addiction (as measures with IAT), and between quality of life (PedsQL^TM^ subscales) and presence of sub-threshold autistic traits (as measured with AQ) in the whole sample (N =145).

		Internet Addiction (IAT)
		rs
Quality of life (PedsQL^TM^ subscales)	Physical functioning	−406 **
	Emotional functioning	−468 **
	Social functioning	−362 **
	School/work activities	−380 **
	Total score	−550 **
Sub-threshold autistic traits (AQ)		−283 **

** *p* < 0.01.

**Table 4 healthcare-11-00186-t004:** Parameter estimates of the relationship between QoL and IAT moderated by AQ.

Dep: IAT N = 141		B (SE)	95% Wald CI (LLCI–ULCI)	Wald Chi^2^	O.R.
Main effect					
	QoL	−0.001 (0.001)	−0.002–0.001	19.43 **	0.999
	SATs	−0.027 (0.015)	−0.056–0.003	3.21	0.947
Interaction effect					
	QoL by SATs	0.000031 (0.000013)	0.000005–0.000057	5.281 *	1.00

B = unstandardized estimated coefficient; LLCI = Lower Level Confidence Interval; ULCI = Upper Level Confidence Interval;Wald Chi^2^ = Wald Chi-Squared test; O.R. = Odds Ratio ; ** *p* < 0.01; * *p* < 0.05.

## Data Availability

The data presented in this study are available on request from the corresponding author. The data are not publicly available due to privacy reasons.

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
