# Peer review of "The Moderator Effect of Subthreshold Autistic Traits on the Relationship between Quality of Life and Internet Addiction"

_healthcare, 2023, doi:10.3390/healthcare11020186_

Round 1
Reviewer 1 Report
Review: Overall, I found this study rather interesting. I believe it had a noteworthy topic, the manuscript was clearly written and communicated well. However, I have some concerns that mainly relate to the study premise. Please see my detailed comments below.- I feel that the Introduction section did not manage to establish resoning to why sub-threshold autistic traits (SATs) would be a moderating factor in the releationship between Internet addicton and quality of life. I.e., Are individuals with SAT expected to experience lower/better quality of life than non SAT individuals?
- The authors report that the Quality of Life Scale measures the core dimensions of quality of life where 0 = never a problem, 4 = almost always a problem, and "Rough item responses are rescaled into a range from 0 to 100, with 0 meaning the worst and 100 meaning the best levels of QoL". But I didn't quite understand how 100 is best level (better) quality of life, if higher points indicate more problems? Could the authors clarify this.
- I didn't quite follow why it would be important to compare differences between students enrolled in a scientific/economics major and students enrolled in a humanities major? Would one's field of study be expected to influece QoL, Internet addiction, or SAT?
- Would the authors agree that an additional limitation of the study is a relatively small sample size, consisting of mostly females?
Reviewer 2 Report
Study examine links between Autism traits, quality of life, and internet abuse as rated by self-report measures administered to college students. Greater research is needed to examine role of autism spectrum/traits and connections to connects to compulsive behaviors, so I commend the authors for assessing this in the present study. Generally, the study is simple, looking at moderation of traits on relationship between QOL and internet use. Limited conclusions can be drawn as this is cross-sectional. Below are some comments to improve the paper and clarify introduction and discussion sections.
Pg 1
Line 30-32 – clarify the difference between SAT and ASD – folks typically consider ASD to include subthreshold so might be key to distinguish that more clearly here.
Line
QOL – more compelling to lead with the links between QOL and Autism and then QOL and internet or other addictions more than this hasn’t been studied in college students. I would lead with this more so than the unique aspect of this study being use of college students.
While you provide some literature on the links between SAT, QOL, and IA, you don’t provide much reason to suggest the direction of your hypothesis. I would want more reasoning as to yh you think that SAT will moderate QOL and IA. Also, need more compelling reason to focus on college kids.
Pg 3 –
Need to include a statement of IRB approval for study.
Measures:
- Please include validity and reliability data for the measures in the text, including internal consistency.
Results
- Why compare majors in QOL? This wasn’t discussed in the intro as of importance or relevance to the question. If you include this, you want to state why you are analyzing subject type differences.
- Can effect sizes be calculated here, gives the reader as sense of how much SAAT impacts IA scores.
- no mention of mean age, gender, ethinicity?
- Why do they think their data is non-normally distributed? Comment on this.
Discussion
- Reason for female lower QOL on physical – could also mention other possible reasons? Maybe there were more sedentary at home, less active? We also don’t know if this was a pre-existing issue/difference as this is a cross-sectional assessment. You don’t have pre-post data on COVID or vaccination data to draw conclusions. Has this finding been reported in on QOL studies in college students?
- The differences in engineering or humanities QOL might be related to course load? Stress or intensity of course? Any data on stress levels by major. I don’t think you make a compelling argument about future salary as a reason. I also don’t know why you did this comparison. Need rational for this.
- Page 7 line 261. When autism traits are lower (so less autism?), QOL affects IT more….so ….might need to reword this. So, QOL and IA are more strongly correlated what AQ are lower, such that when AQ are higher, QOL is less correlated with IA. So, QOL in those with high AQ is less protective on IA scores.
- Discussion need to suggest how clinical services could use this data, should they be trained to assess for SAT? Are there any interventions that have treated IA in college students and/or folks with ASD? Would be good to mention here.
- I also would like a comment on whether this data supports IA as a disorder?
- Limitations – you don’t mention limitations on generalizability, can’t speak to those with ASD diagnosis, self-report aspects of data, what about predominately female sample, and only one measure of each domain was taken?
Reviewer 3 Report
The study is well-organized. My concern is about the statistical analysis applied in this study,
The authors said “Preliminary analyses of data distribution (Kolmogorov–Smirnov test, Skewness and Kurtosis) shown that most of the study’s 156 variables were not normally distributed. Therefore, non-parametric analyses were applied.” I argued that this study should used parametric analyses instead.
Non-parametric tests can be used usually when the measurements are nominal or ordinal. But the author applied a Generalized Linear Model (GLM) to analysis the linear relationship between variables. This is quite unusual since the authors assume “linear” relationships between variables but measures are ordinal (non-linear). Even the author said the study’s 156 variables (items) were not normally distributed, but the normally distribution should be tested on “scale total scores” level, rather than “single item” level. The unit of analysis is “scale total scores” , not the single item scores.
Besides the authors used “Mann-Whitney tests were preliminary conducted to verify differences between males and females and…”
Actually, ANOVA is considered a robust test against the normality assumption. (see Blanca Mena, M. J. et al., (2017). Non-normal data: Is ANOVA still a valid option?. Psicothema. and Knief et al., 2021 Violating the normality assumption may be the lesser of two evilsBehav Res 53, 2576–2590 (2021). Besides, the robustness of F-test to non-normality has been studied from the 1930s through to the present day.
Round 2
Reviewer 3 Report
The author justified the reasons of applying non-parametric statistical methods. I think the reasons provided by the authors could be accept.